# Large-Scale Solar Wind Phenomena Affecting the Turbulent Cascade Evolution behind the Quasi-Perpendicular Bow Shock

**Liudmila S. Rakhmanova ***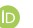**, Maria O. Riazantseva, Georgy N. Zastenker and Yuri I. Yermolaev**

Space Research Institute, Russian Academy of Sciences, 117997 Moscow, Russia
* Correspondence: rakhlud@gmail.com

**Abstract:** The Earth's magnetosphere is permanently influenced by the solar wind. When supersonic and superalfvenic plasma flow interacts with the magnetosphere, the magnetosheath region is formed, which is filled with shocked turbulent plasma. Varying SW parameters influence the mechanisms of formation of this boundary layer, including the dynamics of turbulence behind the bow shock. The effect of the solar wind on the development of turbulence in the magnetosheath was demonstrated recently based on broad statistics of spacecraft measurements. The present study considers the multipoint observations of turbulent fluctuations in the solar wind, in the dayside magnetosheath and at the flanks, to analyze the evolution of the turbulent cascade while the solar wind plasma enters the magnetosheath. Observations of the magnetosheath behind the quasi-perpendicular bow shock are analyzed to exclude the influence of the bow shock topology from consideration. Three basic types of solar wind flows are considered: slow undisturbed solar wind, compressed regions, and interplanetary manifestations of coronal mass ejections. The results show surviving Kolmogorov scaling behind the bow shock for steady solar wind flow and amplification of the compressive fluctuations at the kinetic scales at the magnetosheath flanks for the solar wind associated with compressed plasma streams. During interplanetary manifestations of the coronal mass ejection, the spectra in the dayside magnetosheath substantially deviate from those observed in the solar wind (including the absence of Kolmogorov scaling and steepening at the kinetic scales) and restore at the flanks.

**Keywords:** solar-terrestrial relations; space plasma; turbulence; solar wind; magnetosheath



## 1. Introduction

The Earth's magnetosphere stays permanently under the influence of the solar wind (SW). A wide set of measurements in the interplanetary space during the last century gave general information on the properties of the solar wind and its variability [1,2]. However, the picture of the magnetosphere response on varying SW flow stays still incomplete and presents a challenging problem in a framework of space weather predictions.

The complex structure of the magnetosphere includes the outer boundary layers like magnetosheath (MSH), which forms when supersonic SW plasma faces the magnetosphere. The MSH is located in front of the magnetopause and serves as a link between the SW and the magnetosphere. Thus, processes taking place there must be kept in mind for reliable predictions of solar-terrestrial relations. Description of the plasma flow around the Earth's magnetosphere has been a challenging problem since the early space era (e.g., [3]). Global gas- and magnetohydrodynamic (MHD) description of near-Earth plasma flow helped to obtain satisfying mean plasma and magnetic field parameters in the MSH (e.g., [4–6]). However, the models failed to reproduce fast and spontaneous fluctuations of the MSH parameters, which are widespread behind the bow shock (BS) (e.g., [7,8]). This discrepancy arises from wave processes [9], instabilities, and transients which usually have kinetic nature and can be altogether incorporated into the cascade of turbulent fluctuations [10]. Modern hybrid models of the near-Earth plasma reveal features of the fluctuations in the

boundary layers, which resemble the in-situ measurements (e.g., [11–13]) and demonstrate the important role of kinetic processes and turbulence in the MSH.

To date, mean properties of plasma turbulence in the SW and the MSH, such as characteristics of the magnetic field as well as ion density (or ion flux) fluctuation spectra, are generally known. Usually, the spectra are characterized by power laws ~$f^{-\alpha}$ with $\alpha$ changing at some characteristic plasma scales that form spectral breaks. The scales of particular interest are those around the ion spectral break (occurred at scales close to ion Larmor radius) below which ions are no longer frozen into the magnetic field, and kinetic effects arise. The power law of $-5/3$ at the inertial range of the cascade (at which the MHD description of plasma is usually valid) was suggested by Kolmogorov for turbulent flows [14] and further extended to the turbulent plasmas [15]. At frequencies higher than the ion break (kinetic scales), theories usually predict spectra with a $-7/3$ power exponent (e.g., [16]). Steeper spectra with $-8/3$ exponent at the kinetic scales may be the result of the formation of small-scale structures (e.g., [17]). Comprehensive experimental studies of the turbulence features in the MSH at the scales around and below the proton gyroradius up to electron scales have been performed since the launch of the Cluster mission (see reviews [18,19]). Recent measurements by MMS contributed to a deeper understanding of processes at sub-ion and electron scales (e.g., [20–22]). Experiments show that generally, spectra of the magnetic field, as well as ion flux fluctuations in the MSH, follow a power law with $-5/3$ exponent at frequencies lower than the ion spectral break and with $-(2.8–2.9)$ power exponent at subion scales [23–26]. These properties are close to what is observed typically in the pristine SW without large-scale disturbances [27–31]. Though the distribution of the power law index exhibits maximum at values $-(2.8–2.9)$, other values varying from $-3.5$ to $-2$ occur in the experiments as well [19]. Moreover, the presence of coherent structures or instabilities contributes to the observed spectrum shape, which can result in various spectral features (e.g., [32]).

Unlike the SW turbulence, which can develop freely for steady background, inside the MSH, turbulence always develops in the space confined by the BS and the magnetopause. The presence of boundaries and the distance to them may influence the cascade. Case studies demonstrated that boundaries increased turbulence anisotropy [33,34] as well as their influence on the level of intermittency [34,35]. In addition, the interaction of the SW with the BS may destroy the inertial range of the turbulent cascade [36].

Properties of the MSH fluctuations are highly dependent on the angle $\theta_{BN}$ between the BS normal and interplanetary magnetic field (IMF) vector. Typically, the power of plasma and magnetic field fluctuations are 2–3 times higher behind the quasi-parallel BS compared with the quasi-perpendicular BS; moreover, behind the quasi-parallel BS, the variations of the parameters may be as high as a mean value of the parameter [37]. In addition, these two regimes are characterized by differences in turbulence features [22,38–40]. MSH behind the quasi-perpendicular BS is typically characterized by an $f^{-1}$ spectrum at the frequencies lower than the ion cyclotron frequency [22,38], which is attributed to a superposition of incoherent waves [41] and refers to the absence of an inertial range of the turbulent cascade. On the other hand, Kolmogorov scaling can usually be found behind the quasi-parallel BS [22,38]. At the kinetic scales, the BS topology seems not to influence the magnetic field fluctuation spectrum [22,38], and it is characterized by a power exponent between $-3$ and $-2$ [23,24,26,38]. However, statistical studies of ion flux fluctuations (which mostly represent the fluctuations of density, see the discussion in Section 2.1) demonstrated steeper spectra with a $-3.8$ exponent behind the quasi-parallel BS, while the exponent for the quasi-perpendicular BS was around $-3$ [40]. The nature of this difference is still not clear.

Behind the quasi-perpendicular BS, plasma is dominated by instabilities and low-frequency waves, which arise due to high-temperature anisotropy. The wave mode depends on the value of parameter $\beta p = P_T/P_B$, where $P_T$ and $P_B$ are the ion thermal and magnetic pressures, respectively. For $\beta \leq 1$, Alfven Ion Cyclotron waves are excited, while for $\beta \geq 5$, mirror mode waves are typically observed [9]. Comprehensive analyses have demonstrated the signatures and sources of the wave modes in the MSH and their influence on the spectral

features of the fluctuations [42–45]. In addition, MSH turbulence often demonstrates the presence of Alfven vortices [46,47] at ion scales, which have been detected recently in the SW as well [32,48,49].

Last year, statistical studies showed that features of the MSH turbulence varied with the position of the observation point with respect to the subsolar region and the BS [36,50–52]. Namely, violation of Kolmogorov's $-5/3$ scaling often occurs close to the subsolar regions with its reconstruction at the flanks and in the vicinity of the magnetopause. Statistical results [53] showed a general steepening of the spectra behind the BS at the kinetic scales from $f^{-3}$ for southward IMF to $f^{-3.8}$ for northward IMF. Additionally, the authors showed typical shapes of spectra of compressive fluctuations behind the BS for highly variated SW plasma density, while for steady SW, the spectra often exhibited deviation from Kolmogorov scaling. On the other hand, statistical study [52] presented no dependencies of the MSH turbulence on the variations of the SW parameters. Thus, despite the ability of the large dataset of in situ measurements, the picture of turbulence development behind the BS for changing properties of the SW is still incomplete.

The great difficulty in describing the MSH turbulence evolution with the help of single point spacecraft data is the constant presence of several factors affecting it. Development of the turbulent cascade after interaction with the BS is superimposed by the changes in the upstream SW that affect the MSH processes directly (by changes of the local plasma parameters) as well as indirectly by change of the dynamics of the MSH as the system (i.e., fast movements of the BS position, etc.).

Attempts to correlate the primary properties of the SW plasma and IMF with the properties of the MSH turbulence gave no significant results [52,53]. However, the parameters of the SW can be considered complex depending on the type of the SW. Large-scale SW streams are known to have specific properties depending on their source at the Sun. Most of the time, slow undisturbed solar wind is observed, which originates in the coronal streamers. Coronal mass ejections (CMEs) at the Sun result in the spreading of fast large-scale structures in the interplanetary medium (interplanetary CME–ICME), which can be observed at the Earth's orbit, e.g., in the form of magnetic clouds (MC). At the same time, such structures propagate from the Sun like a piston and compressed plasma forms in front of them, which may be bounded by an interplanetary shock (so-called Sheath region). In addition, similar compressed regions are formed in front of high-speed streams, which originate in the coronal holes (so-called corotating interaction regions (CIRs). A detailed description of the solar wind of different large-scale types can be found in [1].

Recent studies revealed different fundamental properties of the SW plasma associated with different sources at the Sun. Extent study [54] showed substantial differences in anisotropy of magnetic field fluctuations as well as velocity fluctuations within the inertial range of the turbulent cascade for the SW originating in the coronal holes and for sector-reversal plasma regions (which includes heliospheric plasma sheet). The spectral indices also demonstrated slightly different values for several considered types of SW, with the plasma from coronal holes having the scaling of the magnetic field spectra closer to $-5/3$ Kolmogorov scaling than others. At the kinetic scales [55], in the frequency range between the ion break and up to 8 Hz, the $-2.8$ index is typical for slow undisturbed SW, while for the compressed plasma flows, this index usually has higher absolute values of $-3.2$. Moreover, the study demonstrated that for the compressed SW flows, the ion spectral break occurs at scales of proton gyroradius, while for slow undisturbed SW, the break cannot be connected to the single characteristic scale for all cases. Recently effects of the large-scale SW disturbances on the small-scale properties inside the MSH were reported. The occurrence of high-speed jets inside the MSH was shown to be different for periods of interplanetary CMEs and stream interaction regions [56]. Compressed SW flows were shown to be followed by a significant deviation of MHD-scale spectra slope from Kolmogorov $-5/3$ scaling behind the BS [19]. A case study of the ICME passage through the BS and MSH showed that when the compressed Sheath stream interacts with the MSH, some of the MSH small-scale fluctuations can preserve from the SW while some of them

originate inside the MSH [57]. Statistical studies showed differences in magnetosphere response to the various large-scale solar wind phenomena (e.g., [1]), i.e., their different geoeffectiveness. Thus, the interaction of the SW of different types with the BS is likely to result in specific properties of the turbulent cascade throughout the MSH.

The present study aims to describe differences in the development of the turbulent cascade in the MSH during SW streams with specific properties. The study is based on several cases of simultaneous plasma observation in three points of the near-Earth's space: in the SW, in the dayside MSH and at the MSH flank. The study focuses on the range of scales around the ion spectral break. Measurements of the magnetic field from WIND and Themis and measurements of ion flux from Spektr-R spacecraft have been used with time resolution high enough to observe ion spectral break and subion scales. The solar wind flows with three different large-scale properties are considered: slow undisturbed SW, compressed regions, and the interplanetary manifestation of CME. A detailed description of data and methods, together with an example of the analysis, are presented in Section 2, results of the analysis for different SW types are presented in Section 3, obtained statistics are presented in Section 4 and discussed and summarized in Section 5.

## 2. Data and Methods

### 2.1. Spacecraft Measurements Used

To directly consider the spatial evolution of the turbulence properties, simultaneous measurements are required at least at three points of the near-Earth's space: upstream of the BS, downstream of the BS in the subsolar region, and at the flank MSH. For the present study, WIND, Themis, and Spektr-R measurements have been adopted. WIND (WIND NASA homepage https://wind.nasa.gov/, accessed on 1 August 2022) continually monitors the SW at the L1 Lagrange point. Magnetic field measurements are available constantly at WIND with enough time resolution, which makes its data preferable for the current study rather than other SW monitors. Spektr-R [58] has a highly elliptical orbit and, depending on the season, provides measurements both in the subsolar MSH and at the flanks. The Themis\Artemis [59] mission consists of 5 spacecraft, 2 of which (Themis-B,-C or Artemis-1/-2) have been in the Moon's orbit since 2010 and thus cross the MSH at the distant flanks ($X_{GSE} \sim -50$ $R_E$) while 3 others (Themis-A/-D/-E) have seasonally changing near-Earth orbits. It is possible to obtain measurements in the dayside MSH. According to statistical study [25], the ion spectral break typically occurs at frequencies lower than 2 Hz in the MSH. The scales of interest are around ion scales. Thus, measurements with a cadence of 4 Hz or better are required. MFI [60] measurements of the IMF onboard the WIND spacecraft are usually available with an 11 Hz cadence, and FGM [61] magnetic field vector measurements on board Themis spacecraft are available continually with a 4 Hz cadence or better. Magnetic field measurements are not available at the Spektr-R spacecraft. However, the BMSW instrument [62,63] on board the Spektr-R provides continuous time series of unique ion flux vector measurements with a 32 Hz cadence. Fluctuations of ion flux values represent mostly the fluctuations of ion density. The similarity of spectra of ion flux value and density was shown directly by the SW measurements [64,65].

Ion flux (i.e., ion density fluctuations) are compressive fluctuations as well as the fluctuations of the magnetic field magnitude. The compressive component of the fluctuations is usually considered as passively mixed by self-contained Alfvenic turbulence [16], which would result in similar scaling of these two components with differences in fluctuations' power depending on the considered scales. Distinct case studies showed similarities in the shape of density and magnetic field magnitude fluctuation spectra [39,66,67]. A recent case study of MMS measurements also shows the similar shape of spectra of the normalized magnetic field magnitude and normalized density fluctuations [68]. The present study considers only shapes of the spectra (i.e., power exponents, presence of knees, etc.) without preparing any quantitative conclusions on energy exchange rates, etc. Thus, we suggest that a comparison between normalized fluctuation spectra of the ion flux value and magnetic field magnitude is appropriate for the aims of the study.

Plasma measurements at all three points were used to correlate the data time series. SWE data with 92 s time resolution and 3DP data with 3 s time resolution were used on board the WIND spacecraft. SWE [69] measurements were used for the determination of plasma parameters, while 3DP [70] measurements (when available) were adopted for correlation analysis as their time resolution was close to that of the Themis and Spektr-R data. On board the Themis, plasma measurements with 3 s (or 4 s, depending on date and spacecraft) time resolution was provided by ESA [71] instrument, and onboard moments were used. WIND and THEMIS data were adopted from cdaweb.nasa.gov/web-source. Spektr-R data are available at http://catalog-sw-msh.plasma-f.cosmos.ru/ (accessed on 20 November 2022); data with the highest time resolution are available on request; note that data with a high time resolution are available for 10% of the period of Spektr-R operation (2011–2019) due to telemetry limitations.

### 2.2. Tracking of the Plasma Volume

To compare properties of fluctuations at distant points, one has to find observations of the same plasma volume at these points first. In the present study, correlation analysis of plasma measurements at three considered spacecraft was adopted for these purposes. First, initial propagation time was determined via the distance between the spacecraft pairs and the mean plasma velocity measured at them. Second, the data series of ion density measurements were linearly interpolated to a common time grid to have an identical time resolution of 3 s. Then, the correlation coefficient was calculated for a set of time shifts between two data series. The set of time shifts was chosen to lie in $+-30$ min intervals from the initial propagation time. Such an assumption seems to be reasonable for distances of about ~200 $R_E$ because it eliminates unphysically random matches in data rows and, on the other hand, accounts for differences in propagation times of different small-scale plasma structures. Then, the shift which corresponded to the maximum correlation coefficient was chosen as the first approximation of the real propagation time. Note that in the case of WIND (SWE) data with original 92 sec time resolutions, this procedure results in a significant enlargement of the number of data points in time rows. However, the further analysis includes manual checking of the shifts, so the mathematical unreliability resulting from this enlargement becomes insignificant.

The main problem in matching time series from several spacecraft is the difference in propagation speed (and consequently in propagation times) of different small-scale structures and discontinuities on their edges. This difference may be insignificant at the scales of ~20 $R_E$, as was shown in [72]. However, it may result in significant uncertainties when comparing data from 200 $R_E$-separated points. Such a difference usually results in discrepancies between data rows when considering smaller (~10 min) time intervals. The problem is illustrated in Figure 1. The figure shows density measurements by WIND (3DP) and Spektr-R over 7 h (a) and 2 h (b) intervals, respectively, on 14 April 2013. Spacecraft positions are shown in panel (d). One can see good large-scale matching of time profiles of density measurements which suggests that both spacecraft measure the same plasma. The discrepancy between the interplanetary shock registration times at ~23:00 results from its significantly higher propagation speed (~480 km/s according to http://ipshocks.fi catalog (accessed on 20 November 2022)) compared with the ambient plasma (370 km/s). At smaller scales (panel b), one can find several examples of structures (discontinuities, current sheets, etc.) that match each other (highlighted by boxes). On the other hand, there are a lot of similar small-scale features which do not match in time for the selected shift between the spacecraft data rows. This may result from either different speed of propagation of small-scale structures or their fast time evolution. The latter option is the object of interest of the current study as it reflects the evolution of small-scale variations incorporated into the turbulent flow.

Thus, the correlation analysis is useful when considering two closely located spacecraft (e.g., Themis–Spektr-R pair) and to obtain the first approximation of the propagation times. However, when considering two distant points and focusing on small scales, one has

to choose the final time lag manually. For the current study, the time lag was chosen to match the maximum amount of the small-scale plasma structures in the considered ~30 min intervals.

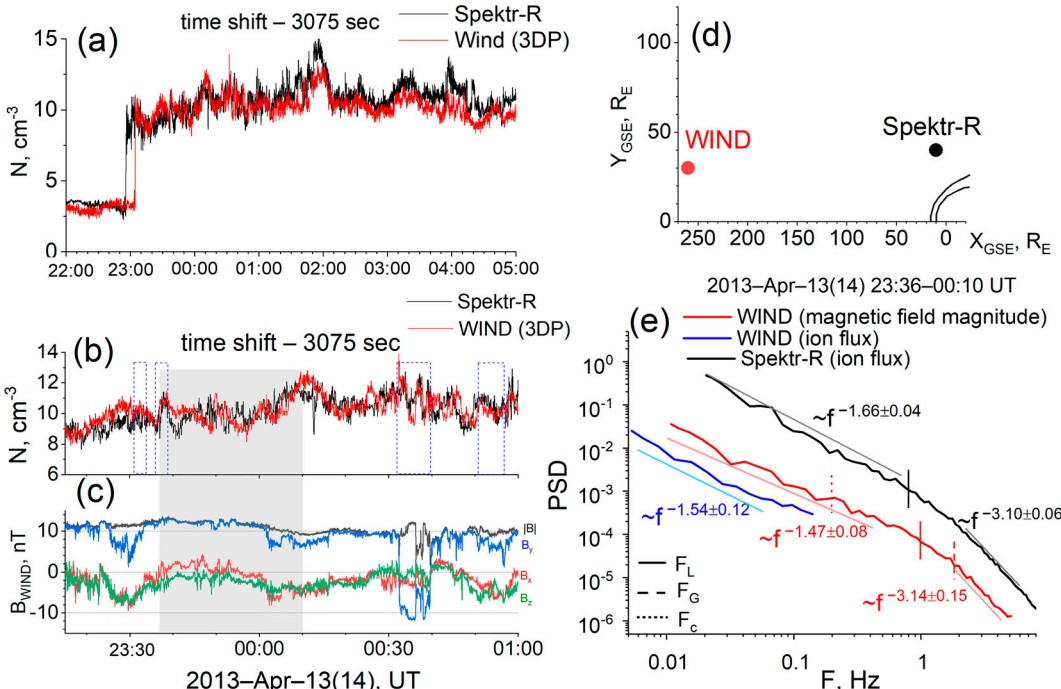

**Figure 1.** (**a**,**b**)—density measurements at Spektr-R (black line) and WIND 3DP (red line) on 14 April 2013; WIND data are shifted in time by 3075 sec; (**c**) WIND magnetic field components, shifted by the same time; (**d**) s/c positions for the analyzed case; (**e**) comparison of the normalized magnetic field magnitude (red line) and ion flux (blue line) spectra at WIND and ion flux spectrum at Spektr-R (black line) for the analyzed case.

*2.3. Fourier Analysis*

The study uses the Fast Fourier transform (FFT) to quantify the properties of turbulent fluctuation spectra. The FFT has been performed for intervals of ~17 min or ~34 min duration (when available), which is the compromise between the required number of data points and the constancy of the ambient conditions. An example of the interval used for the FFT (23:36–00:10 UT in Spektr-R time) is presented by the shaded area in Figure 1b. Figure 1c presents magnetic field components and magnitude measured at WIND and shifted in time by propagation time between WIND and Spektr-R calculated as described above. One can see that there are no significant rotations of the field or any sudden significant changes in density during the chosen interval. To compare fluctuations of different parameters (magnetic field magnitude and ion flux), the measurements were normalized to the mean value of the parameter over the interval used for the spectrum calculation. Fourier spectra were averaged in the frequency domain with the Hamming window. Comparison of the resulting spectra from WIND (red line) and Spektr-R (black line) for the interval is presented in Figure 1e. The spectra here and further in the text are the Power Spectral Densities (PSD) of the value $dX/<X>$, where $dX$ is the variation of the parameter (magnetic field magnitude or ion flux) and $<X>$ is the parameter's mean value over the considered interval. Vertical lines at the spectra denote characteristic plasma frequencies: ion gyrofrequency Fc, Taylor-shifted ion gyroradius $F_G = V/2\pi\varrho$ and Taylor-shifted ion inertial length $F_L = V/2\pi L$. In addition, a spectrum of ion flux measurements by WIND (3DP) was added to the figure (blue line). The 3DP measurements have a low cadence, and the interval for FFT has been enlarged to 23:19–00:27 UT in the Spektr-R time. Linear parts of the spectra were approximated with power laws. Lines in Figure 1e and in the text denote the frequency ranges used for the approximation procedure. Spectra of ion flux

from Spektr-R and magnetic field measurements from WIND demonstrate no clear break but a smooth transition from one power law index to another at frequencies around $F_G$ and $F_L$. This is typical of the spectra of ion flux fluctuations (e.g., [28]).

Note that sometimes peculiarities in spectra are observed, such as plateaus or bumps. When such effects occur close to the lowest considered frequency, the sufficient number of points used for approximation at the MHD scales (at least 10 points) may be unavailable. For such cases, when possible, extended data intervals are used for a spectrum calculation. Otherwise, the approximation is not performed.

### 2.4. Turbulence Evolution between the L1 and the Bow Shock

The example presented in Figure 1 aims to check if one can state that turbulence properties in L1 can be used as a proxy of those in front of the BS. For the presented case, the spacecraft were separated by more than 250 $R_E$; still, the spectra demonstrated similar shapes and features. Taking into account approximation errors, typical Kolmogorov scaling is observed at frequencies lower than the ion spectral break. Note that the low-frequency range of the spectra contains a substantially lower number of points compared with the high-frequency range, which results in larger errors in spectrum approximation. At the subion scales, approximation gives identical (taking into account the estimation errors) values of the spectral slopes (power exponents).

The considered interval was associated with the SW of type Sheath. Such SW streams usually contain a significant number of small-scale structures used in the current example to demonstrate the differences in their propagation speeds. However, a similar comparison (not shown) for undisturbed solar wind (of type "Slow") was prepared and demonstrated similar spectra at the L1 point and close to the BS in the SW as well. Thus, we suggest that the spectra of magnetic field magnitude fluctuations on board the WIND can be successfully compared with the ion flux fluctuation spectra on board the Spektr-R spacecraft. Note that the presented example refers to the insignificant separation of the spacecraft pair in the $YZ_{GSE}$ plane. For large separations (>90 $R_E$), distinct small-scale features of the flow may be captured by one of the spacecraft and not observed at other [73]. This may result in a worse resemblance of the spectra at the spacecraft pair. However, for the present study, the separation in the $YZ_{GSE}$ plane usually lies within 90 RE.

### 2.5. Determination of the BS Type

To eliminate the influence of the BS topology and the foreshock processes on the turbulence evolution from the consideration, only cases of the MSH behind the quasi-perpendicular BS were considered. The selection was performed according to the $\theta_{BN}$ angle value. $\theta_{BN}$ is the angle between the IMF vector and the BS normal in the place where plasma enters the MSH. To calculate the angle, the method described in [37] was used. To determine the point of plasma entrance, the position of the spacecraft in the MSH was traced to the BS along the stream line, calculated with the help of the model [3]. Then IMF parameters were traced from the WIND spacecraft to the entrance point, and the angle was calculated. For the purposes of the current study, only intervals with $\theta_{BN} \geq 45°$ (quasi-perpendicular BS) at all of the MSH spacecraft were chosen.

### 2.6. Data Selection

For the analysis, the database of the BMSW measurements in the MSH was used, and the corresponding positions of the Themis spacecraft were checked. The database of the BMSW measurements in the MSH included ~170 h and was performed in the statistical study [25]. To be chosen for further analysis, the spacecraft pair in the MSH must (1) stay inside the MSH for at least 2 h; (2) be located at the same flank of the MSH, e.g., with the same sign of the $Y_{GSE}$ component and behind the quasi-perpendicular BS; (3) provide fast enough measurements of the magnetic field magnitude/ion flux value; (4) demonstrate similarities in density time series and high value of the correlation coefficient (more than 0.6). Then, for the obtained interval, the WIND data were checked. If there were (1) good

correspondence between Themis–WIND and Spektr-R–WIND density time series, and (2) fast IMF measurements available at WIND spacecraft, the interval was selected for further analysis. The obtained intervals were sorted according to the large-scale SW type determined with the help of the catalog [74], available at http://iki.rssi.ru/pub/omni/catalog/ (accessed on 20 November 2022). The catalog includes the following types of the SW: (1) "Slow" is the slow undisturbed SW; (2) Magnetic cloud ("MC") and (3) "Ejecta" are different interplanetary manifestations of the CMEs distinguished primarily by the behavior of the interplanetary magnetic field which is stronger and more regular for MC [74,75]; (4) "Sheath" is the compressed region in front of MC and Ejecta; (5) "CIR" is the compressed region in front of the fast SW stream from the coronal hole. A detailed description of the SW types can be found in the original paper [74]. The time series were matched, and the interval for the FFT was chosen as it was demonstrated in Section 2.2. Altogether 12 intervals were obtained, which refer to the Slow (3 intervals), Sheath (1 interval), CIR (5 intervals), Ejecta (1 interval), and MC (2 intervals) types of the SW. For the present study, the SW types were grouped into 3 species: (1) undisturbed slow SW–Slow; (2) disturbed compressed SW streams–Sheath, CIR; (3) disturbed SW associated with ICMEs–MC, Ejecta. Note that the obtained amount of each type of SW does not reflect the typical occurrence rate of these types. This is likely to be the result of a low number of cases that satisfy the selection criteria and due to the limited amount of Spektr-R high-resolution data.

### 2.7. Taylor Hypothesis

Single spacecraft provide time measurements in the moving inhomogeneous plasma. The spacecraft speed is much lower than the plasma speed, and the time evolution of plasma parameters may be interpreted as spatial inhomogeneity, which passes over the spacecraft with plasma speed. Usually, obtained frequency spectra are interpreted by adopting the Taylor hypothesis [76]. The hypothesis allows direct recalculation from time scales to dimensional scales if the phase speed of the dominant wave mode is close to zero in the plasma frame. Taylor hypothesis may be used when $V/V_A > 0.3$ ($V_A = H \times (4\pi N)^{-1/2}$ is the Alfven speed) [77,78], which is valid in the great majority of cases in the SW and flank MSH. However, this condition may be ruined in the dayside MSH. For all the analyzed cases (except one when Spektr-R measurements were used in the dayside MSH), this ratio was checked for the spacecraft in the dayside MSH. The values of the $V/V_A$ are listed in Table 1. The minimum observed value is 0.4. Thus the mentioned conditions of [77,78] are satisfied.

**Table 1.** Characteristic background parameters of the analyzed cases.

| № | Date | SW Type | Dayside MSH Parameters | | | SW Parameters | | | | | |
|---|---|---|---|---|---|---|---|---|---|---|---|
| | | | $V/V_A$ | $\beta_p$ | $\alpha(V,B)$, ° | N, cm$^{-3}$ | V, km/s | \|B\|, nT | $T_p$, eV | $\beta_p$ | $\alpha(V,B)$, ° |
| 1 | 2014-02-08 | Ejecta | 0.4 | 0.7 | 75 | 6.1 | 459 | 11.7 | 8.4 | 0.15 | 61 |
| 2 | 2014-02-09 | SLOW | 0.5 | 2.6 | 150 | 4.3 | 431 | 7.0 | 2.9 | 0.10 | 119 |
| 3 | 2014-02-16 | MC | 0.5 | 0.7 | 64 | 9.4 | 405 | 17.0 | 3.0 | 0.04 | 78 |
| 4 | 2014-02-27 | SLOW | 0.8 | 4.3 | 55 | 17.4 | 353 | 4.7 | 3.2 | 1.01 | 121 |
| 5 | 2014-02-27 | CIR | 0.7 | 2.0 | 82 | 21.2 | 472 | 14.3 | 21.3 | 0.89 | 88 |
| 6 | 2014-07-09 | SLOW | – | – | – | 8.4 | 351 | 6.2 | 4.6 | 0.40 | 82 |
| 7 | 2015-03-17 | SHEATH | 1.7 | 3.5 | 151 | 24.2 | 545 | 21.4 | 67.1 | 1.43 | 136 |
| 8 | 2015-03-17 | MC | 0.5 | 1.5 | 84 | 9.7 | 563 | 18.4 | 5.6 | 0.06 | 62 |
| 9 | 2015-07-04 | CIR | 1.2 | 2.2 | 75 | 30.5 | 365 | 14.5 | 7.7 | 0.45 | 87 |
| 10 | 2016-05-21 | CIR | 1.9 | 6.2 | 109 | 10.4 | 493 | 8.9 | 8.6 | 0.46 | 123 |
| 11 | 2017-11-15 | CIR | 1.0 | 3.5 | 81 | 27.9 | 425 | 9.7 | 2.6 | 0.32 | 102 |
| 12 | 2017-12-04 | CIR | 2.2 | 3.8 | 97 | 31.8 | 326 | 4.9 | 2.5 | 1.34 | 82 |

Note that another constraint on adopting the Taylor hypothesis is the presence of whistler waves [78]. However, this wave mode is not widespread in the MSH and SW and can usually be found at scales substantially smaller than considered in the present study (e.g., [79,80]).

## 3. Results

### 3.1. Undisturbed Solar Wind

Figure 2 presents the first considered case observed during steady slow SW of type Slow. Panel (f) presents the location of the spacecraft (the BS and the magnetopause are shown schematically). Themis-D crossed the magnetopause at ~09:30 UT on 27 February 2014 and stayed inside the subsolar MSH till ~16:50 when it entered the SW for 1 h. Spektr-R was scanning the flank MSH till ~17:00 when the coming of the interplanetary shock resulted in the inward motion of the BS, and Spektr-R entered the SW for 40 min. Panels (a–c) present proton density, bulk velocity, and temperature measured by WIND 3DP (red line), Spektr-R (black line), and Themis-D (blue line) during the analyzed case.

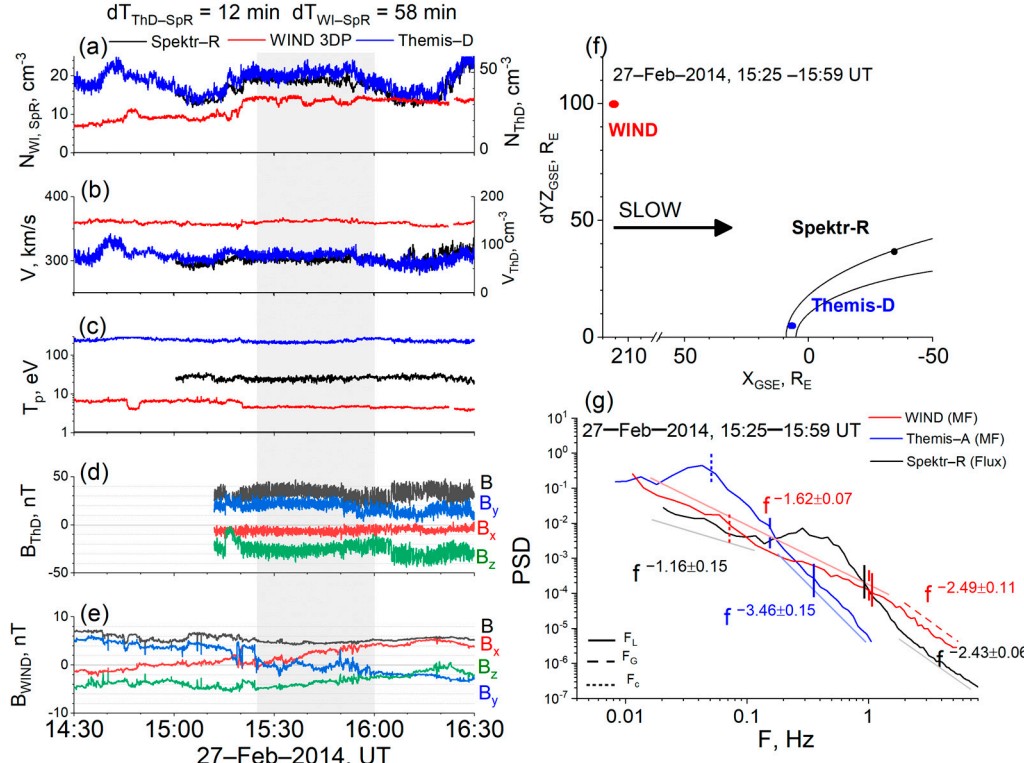

**Figure 2.** Considered interval on 27 February 2014: (**a–c**) Proton density, velocity, and temperature measurements by Spektr-R (black line), WIND 3DP (red line) and Themis-D (blue line), (**d**)—IMF measurements by WIND; (**e**)—IMF measurements by Themis-D; (**f**) spacecraft positions; (**g**) comparison of the normalized Fourier spectra in the SW (red line, magnetic field magnitude fluctuations at WIND), in the dayside MSH (blue line, magnetic field magnitude fluctuations at Themis-D) and at the flank MSH (black line, ion flux fluctuations at Spektr-R) for the analyzed event.

Panels (d) and (e) show magnetic field vectors measured at Themis-D and WIND, respectively. All the spacecraft time series are shifted to match in time with the Spektr-R according to the technique described in Section 2.2. Time shifts are denoted in the figure. The time series of the parameters resemble each other; however, amplitudes of the parameters differ in accordance with the spacecraft positions. The densest, hot, and slowed-down plasma is observed at Themis-D in the dayside MSH. In addition, magnetic field magnitude significantly increases in the dayside MSH compared with the initial IMF. The shaded area denotes the interval (15:25–15:59 UT) used for spectrum calculation. The $\theta_{BN}$ angles for both Spektr-R and Themis-D were 75°, so they are located behind the quasi-

perpendicular BS. Thus, the case compares measurements in the subsolar MSH and at the flank in the vicinity of the BS. Mean characteristic parameters such as the angle between velocity and magnetic field vectors, $\beta_p$, and Taylor ratio for the spacecraft in the subsolar MSH are summarized in line 4 of Table 1; an extended version of the Table with more parameters is presented in the Supplementary Materials.

Figure 2g shows Fourier spectra of magnetic field magnitude fluctuations from WIND (red line) and Themis-D (blue line) and of the ion flux fluctuations from Spektr-R (black line). Mean characteristic plasma scales are denoted for each spectrum by corresponding colors and line types. The spectrum in the undisturbed SW (red line) demonstrates a typical shape with Kolmogorov scaling at frequencies lower than the break and $-2.49 \pm 0.11$ power exponent at higher frequencies. Behind the BS in the dayside MSH (blue line), the bump in the spectrum is formed at a frequency of ~0.04 Hz, which is close to the ion cyclotron frequency. Spectrum at the MHD scales cannot be approximated because of the bump. At the kinetic scales, the spectrum is substantially steeper than in the SW and than observed usually in the SW and in the MSH. Such steep spectra are typical for the presence of coherent structures like Alfven vortices [46,47]. In addition, the steep spectrum may be a continuation of the bump due to instability which is observed at lower frequencies. At the MSH flank, the spectrum is shallower at the MHD scales than observed typically for the developed turbulence in the undisturbed SW characterized by the Kolmogorov scaling. At the kinetic scales, the spectrum is slightly shallower than the corresponding spectrum at WIND. In addition, a bump in the spectrum occurs at frequencies around 0.3 Hz, and it covers the frequency range [0.2–2] Hz. Thus, for the presented case of undisturbed slow SW flow, the turbulent spectrum has typical features in the SW, steepens at the kinetic scales when plasma crosses the BS and recovers its shape at the kinetic scales when the plasma moves toward the flanks, at the MHD scales spectrum at the flank MSH in the vicinity of the BS deviates from Kolmogorov scaling.

Interestingly, the bump in the spectrum seems to survive during plasma propagation through the MSH through its frequency changes. This may be the signature of the local wave process embedded in the plasma flow, with the characteristic scales changing with the background parameters. In the dayside MSH, the parameter $\beta_p$ is around 4. According to [9], this condition is favorable for the formation of mirror instability. In addition, the magnetic field magnitude and ion density vary in antiphase (not shown), which also confirms the suggestion of the mirror-mode nature of the bump. Unfortunately, the type of wave mode responsible for the bump in the flank MSH cannot be identified due to the absence of magnetic field measurements there.

Other cases were analyzed in a similar way. Here and below, spectra obtained in the SW, in the dayside MSH, and at the flank are marked in red, blue, and black, respectively. For 11 of 12 cases, one of the Themis spacecraft was in the dayside MSH, and Spektr-R was downstream from it at the flank. For one of the cases (9 July 2014), Spektr-R was in the dayside MSH, and Themis spacecraft was at the flank. For each spectrum, characteristic frequencies are denoted at the spectrum. Magnetic field measurements are not available for the Spektr-R, so only inertial length and corresponding frequency could be calculated for this spacecraft. The full set of parameters of the background plasma is summarized in the Table S1 of Supplementary Materials.

Two more examples of the spectra changes in the MSH for Slow SW are presented in Figure 3. For the case shown in the left panel of Figure 3 (9 February 2014), the selected interval was 04:31–04:48 UT. Spektr-R was close to the magnetopause at the flank MSH, while Themis-D was in the dayside MSH in the vicinity of the magnetopause. In the SW at the MHD scales, the spectrum follows the $f^{-1.4 \pm 0.3}$ power law. Though the spectrum is somewhat flatter than the $f^{-5/3}$ law, it still has Kolmogorov scaling taking into account the errors. At the kinetic scales, the spectrum follows $\sim f^{-7/3}$ power law, which is shallower than typically observed in near-Earth plasma, however similar to the theories' predictions (e.g., [16]). Interestingly, the spectrum in the SW demonstrates a plateau (or knee) at the transition scales between MHD and kinetic regimes. Such a plateau is usually observed

for density spectra in the SW [81,82]. Though this plateau is not typical for magnetic field magnitude spectra in the SW, some of the case studies present similar spectral features of compressive components [23] or for the spectral component which corresponds to waves and coherent structures [32]. In the dayside MSH, according to the Themis-D measurements, the spectrum has a typical shape with Kolmogorov scaling at the MHD scales and ~−2.9 slope at the kinetic scales, which is close to mean values for SW and MSH, but steeper than in the SW for the current case. At the flank, the ion flux fluctuation spectrum obtained by Spektr-R is characterized by slopes ~−1.8 and ~−2.6 at the MHD and kinetic scales, respectively. Thus, at the kinetic scales, the spectrum has a slope close to −8/3 obtained in the theories (see Introduction).

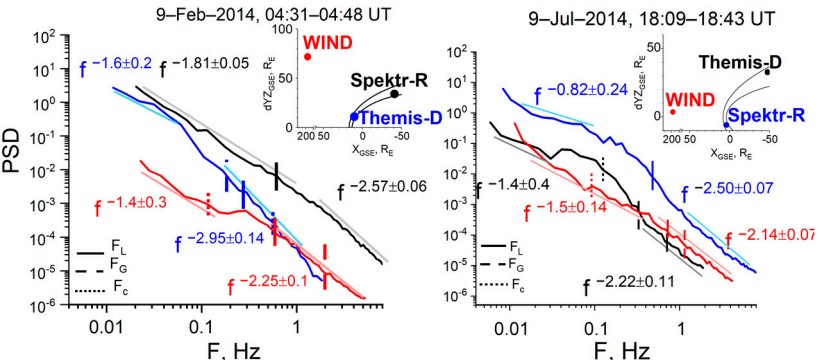

**Figure 3.** Comparison of the normalized spectra in the SW (red line), in the dayside (blue line) and flank (black line) MSH for the undisturbed SW flow of type Slow for the cases on 9 February 2014 (**left**) and on 9 July 2014 (**right**).

The third analyzed case occurred on 9 July 2014 (right panel of Figure 3). Spektr-R was located in the dayside MSH in the vicinity of the BS, and Themis-C was located in the flank MSH close to the magnetopause during 18:09–18:43 UT. In the SW, the spectrum of magnetic field magnitude fluctuations shows Kolmogorov-like scaling at the MHD scales and is flatter than the typical spectrum at the kinetic scales with a slope of ~−2.14. Behind the BS (Spektr-R, blue line for this case), the spectrum is shallower at the MHD scales and has a slope of −2.5 at the kinetic scales, which is steeper than in the SW. At the flank, the magnetic field magnitude spectrum has nearly Kolmogorov scaling at the MHD scales and a slope **of** −2.22 at the kinetic scales, which is close to the slope in the SW. Inside the MSH, both spectra exhibit a bump at the transition scales, which is more pronounced in the magnetic field magnitude spectrum by Themis-C. At Themis-C, the bump is likely to be at a frequency close to proton cyclotron frequency.

Thus, for the steady slow SW flow, the spectrum of turbulent fluctuations changes slightly in the MSH, with the effects decreasing with the distance from the BS both in the dayside MSH and at the flank. The changes in spectra usually include flattening at the MHD scales and steepening at the kinetic scales at the dayside MSH in the vicinity of the BS and recovery of the spectrum properties in the vicinity of the magnetopause and the flanks. This result corresponds well with the previous statistical findings [50,51]. However, the study of the magnetic field fluctuation spectra in the dayside MSH did not present Kolmogorov scaling throughout the dayside MSH [36]. This may be due to the differences in the evolution of the compressive and incompressive components of the cascade throughout the MSH.

### 3.2. Disturbed Compressed SW Flow

Figure 4 presents an example of the analysis of the compressed plasma interaction with the MSH on 4 July 2015. The left panel presents the spacecraft positions, and the panel on the right presents the comparison of the corresponding spectra. For the considered case, the CIR solar wind type was observed. During the analyzed interval 14:05–14:39,

Themis-E was in the central part of the dayside MSH, and Spektr-R was in the center of the flank MSH several $R_E$ downstream from Themis-E. For both spacecraft, $\theta_{BN}$ angles were estimated as 75°.

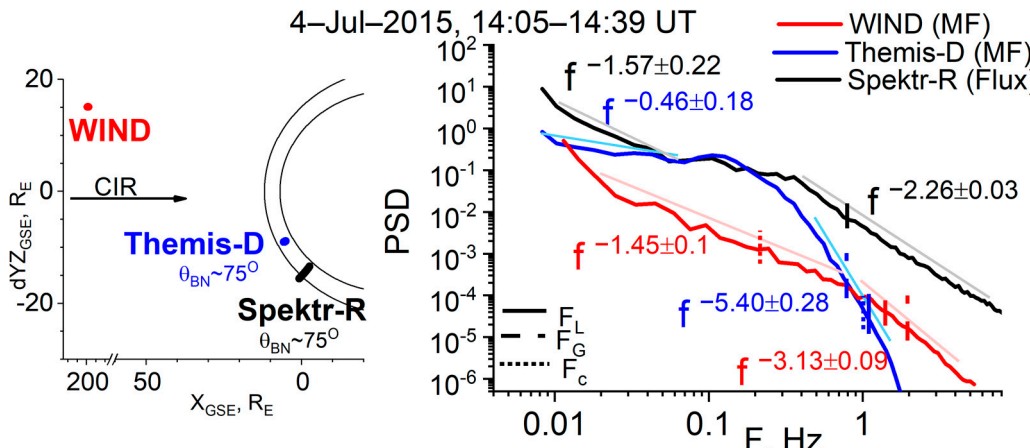

**Figure 4.** (**Left**)—spacecraft positions; (**right**)—comparison of the normalized Fourier spectra for the SW of type CIR.

For the presented case, the spectrum in the SW is slightly shallower than Kolmogorov's at the MHD scales and steeper than typical scaling at the kinetic scales with a −3.13 slope, which is often observed for the compressed SW streams [55]. Behind the BS, the spectrum substantially flattens at the MHD scales and becomes clearly non-Kolmogorov. At the transition scales, a bump in the spectrum is observed. Note that the bump is observed at frequencies significantly lower than any of the characteristic ion frequencies, which implies that this is not a result of any specific wave process. At the kinetic scales, untypical steepening of the spectrum occurs, with the slope being ~−5.4. The slope modulus is significantly larger than in the SW. Downstream from Themis-E, the Spektr-R registered spectrum with Kolmogorov scaling at the MHD scales, a plateau at the transition scales, and ~−7/3 power exponent at the kinetic scales.

Note that the slope −5.4 is highly untypical for the ion kinetic scales in the MSH though it is often observed at smaller electron scales at which further steepening occurs (statistics can be found in [24]). Additional analysis of the magnetic field magnitude fluctuation spectra in the MSH by Themis (not shown here) was performed to check if the untypically steep spectra are the results of the wrong data processing. More than 20 examples of spectra from different Themis spacecraft from various years and spacecraft positions were considered. The analysis demonstrated that such steep spectra sometimes could be found during periods of high magnetic field magnitude (>50 nT, similar to what was observed for the case shown in Figure 4), and there was no evidence of the artificial nature of such a high absolute value of the slopes. In addition, a statistical study [65] presented slopes close to −4.5 for the spectra of ion flux fluctuations downstream from the interplanetary shocks. Thus, the presence of crucially steep spectra in the dayside MSH is likely to be due to the compression typical for CIRs and amplified in the dayside MSH. Note that similarly steep spectra were predicted, e.g., for a cascade of critically balanced Alfven waves parallel to the magnetic field [83]. Observation of such a steep spectrum in the dayside MSH for disturbed SW may be the result of specific conditions (substantial draping of the magnetic field in front of the magnetopause etc.) favorable for cascade development parallel to the magnetic field.

Altogether 6 cases of turbulent cascade evolution for SW of types CIR and Sheath were considered. Interestingly, for all of the cases, the fractional distance (which denotes the position of the spacecraft with respect to the BS and magnetopause) varies; however, no dependence of the spectra parameters on the distance can be found. For this reason, only one of the cases is presented here, while the parameters of others are presented in Section 4.

All other cases share the features of the case presented in Figure 4. The cases demonstrate Kolmogorov scaling at the MHD scales in the SW and further substantial deviation from the Kolmogorov scaling behind the BS in the dayside MSH. At the flank, MSH spectra exhibit clear Kolmogorov scaling except for a single case on 21 May 2016 when the Spektr-R spacecraft was close to the dayside region and in the vicinity of the BS; for this case, the spectrum is slightly shallower than Kolmogorov's that is typical for region closer to the dayside MSH [36,40].

At the flank, the spectral slope at the kinetic scales is usually close to $-7/3$. Thus, at the flank, spectra are flatter than typically observed in the undisturbed SW and MSH plasma.

### 3.3. Disturbed SW Flow Associated with ICMEs

The analysis includes 3 cases of the SW associated with ICMEs: two of them refer to MCs, and one is classified as Ejecta. Figure 5 demonstrates (similar to Figure 4) the results of analysis for one of the cases on 16 February 2014 for the SW associated with MC. Themis-E was in the central part of the dayside MSH, and Spektr-R was at the flank, far enough both from the BS and the magnetopause. The spectrum in the SW is somewhat steeper than Kolmogorov's at the MHD scales with the slope $-1.89 \pm 0.13$ and follows the $f^{-2.5}$ law at the kinetic scales. The spectrum steeper than Kolmogorov's at the MHD scales may be a result of discontinuity that occurred during the interval or a display of turbulence anisotropy [84]. As visual inspection does not show any substantial discontinuity during the interval, the second explanation is more favorable. In the dayside MSH, the spectrum is shallower than Kolmogorov's, with a ~$-0.7$ slope at the MHD scales; at the kinetic scales, the spectrum steepens significantly compared with the one in the SW and has a slope of ~$-4.3$. Such changes in spectrum in the dayside MSH may be the result of a network of Alfven vortices [47]. At the flank, the spectrum has clear Kolmogorov scaling at the MHD scales and is characterized by a $-2.5$ slope at the kinetic scales, which is similar to the spectrum in the SW.

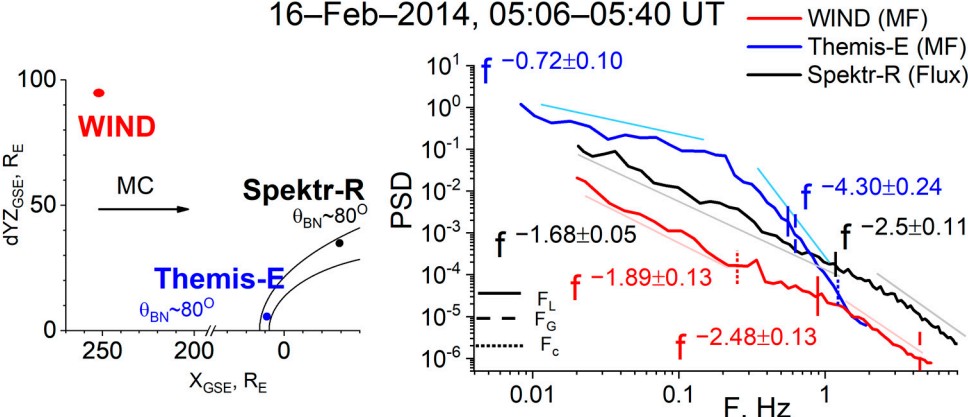

**Figure 5.** (**Left**)—spacecraft positions; (**right**)—comparison of the normalized Fourier spectra for the SW of type MC.

Two other cases share the same features at the kinetic scales: in the dayside MSH the steepening is observed with the further restoration of the spectrum to the values observed in the SW. There is no tendency for untypically flat spectra at the MSH flanks. There are some differences in spectra evolution at the MHD scales. For one of the cases, significant flattening occurs in the dayside MSH with the further restoration of the Kolmogorov scaling at the flank, similar to the example presented in Figure 4. However, for another case (17 March 2015, line 8 in Table 1), deviation from the Kolmogorov scaling is observed both in the dayside MSH and at the flank. For this case, Spektr-R was probing plasma in the vicinity of the BS, which can result in a significant deviation from the Kolmogorov scaling.

## 4. Relation between the Spectra Properties in Different Regions of the Near-Earth Space

Twelve obtained cases give a possibility to reveal differences in turbulence evolution in the MSH behind the quasi-perpendicular BS for steady slow SW, for disturbed compressed SW flows, and for SW associated with ICMEs. The resulting spectra slopes in the dayside and flank MSH versus the slopes of the spectra in the SW are presented in Figure 6, where colors denote different SW types. Note that for two cases, the MHD part of the spectrum cannot be reliably approximated for dayside MSH for the lack of data points in the linear part of the spectrum; the same problem arises for another two cases in the SW. This results in a reduced amount of data points in Figure 6a,b.

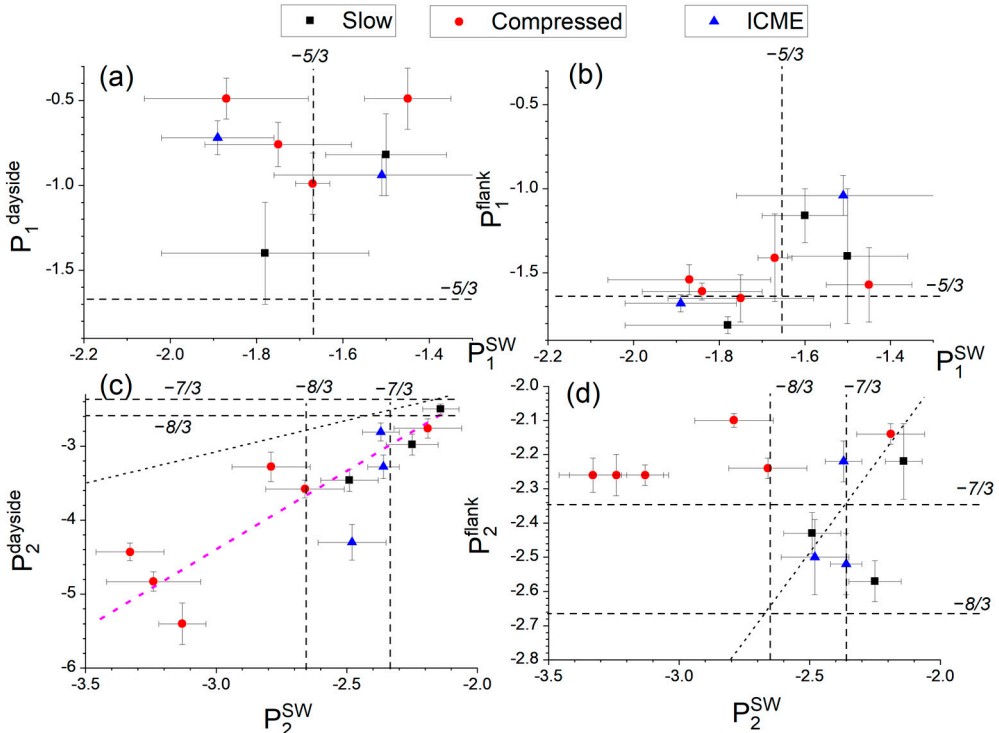

**Figure 6.** Spectral slopes at the MHD scales (**a**,**b**) and at the kinetic scales (**c**,**d**) at the dayside MSH vs. the SW (**a**,**c**) and at the flank MSH vs. the SW (**b**,**d**); colors denote different large-scale SW type: Slow (black), ICME (blue), Compressed (red).

Figure 6a demonstrates that there is no dependence of the MHD-scale slope ($P_1$) downstream of the BS from its value in the upstream SW. However, the Kolmogorov scaling may be observed in the dayside MSH during the slow undisturbed SW. For the disturbed SW of all kinds, the spectra in the dayside MSH have substantially non-Kolmogorov scaling. This result corresponds well to the mean properties of the magnetic field fluctuation spectra behind the BS and in the dayside MSH (e.g., [36,40]). Note that in the study [65], no signature of the spectra flattening downstream of interplanetary shocks at the MHD scales was observed. That assumes the differences in shock effect on the freely propagating plasma and plasma behind the standing BS.

According to Figure 6b, the Kolmogorov scaling generally restores at the MSH flanks in agreement with the statistics for magnetic field fluctuations [36]. There are two exceptions with the slopes $P_1 = -1$ for the SW associated with ICME and $P_1 = -1.14$ for the Slow SW, which are observed in the very close vicinity of the BS and are discussed in Section 3.

Figure 6c demonstrates clear dependence of the kinetic-scale slope ($P_2$) in the dayside MSH on the same slope in the SW. The linear fit of the dependence is denoted in the figure by the dotted magenta line. Pearson's correlation coefficient for this dependence is 0.85. The black dotted line denotes equal values of the slopes in the upstream and downstream regions. Clear steepening of the spectra is observed in the dayside MSH for

all SW types. In addition, while Slow SW and ICMEs exhibit spectra that corresponds to the predictions of theories (with $-7/3$ or $-8/3$ slopes at the kinetic scales), the compressed SW streams are usually characterized by steeper spectra in the SW and untypically steep spectra downstream of the BS.

Previous statistical studies [50,51] demonstrated generally steeper spectra in the MSH in the vicinity of the BS than those typically observed in the SW. The present study also demonstrates a clear steepening of the spectra in the dayside MSH compared with those obtained simultaneously in the SW. Moreover, steepening of the spectra in the downstream regions of the interplanetary shocks was demonstrated [65] for the ion flux fluctuation spectra. However, statistics of the magnetic field fluctuations [36,52] did not show any peculiarities in spectral slope values in this region. However, here and in [50,51], the compressive component of the fluctuations was considered, while studies [36,52] focused on the incompressive magnetic field fluctuations. Thus, the obtained discrepancy may be a signature of differences in the effect of the BS on cascades of compressive and incompressive fluctuations.

Figure 6d shows slope $P_2$ at the flank MSH versus the corresponding slope in the SW. The dotted line denotes equal values of the slopes in both regions. There is no clear dependency. However, several features of specific SW streams can be seen. For the Slow SW and ICMEs, the slopes in the SW and at the MSH flank have similar values close to $-8/3$ and $-7/3$. For the compressed SW plasma, the slope values do not depend on those in the SW and are substantially increased to the values $|P_2| < 7/3$. As far as we are aware, this feature of the MSH turbulence was not reported previously based on experimental studies.

## 5. Discussion and Summary

The cases considered above suggest specific features in the evolution of turbulence at the BS and the MSH for different large-scale SW streams. For the undisturbed slow SW, the turbulent spectra may be slightly modified at the BS. The modification may include the deviation from the Kolmogorov scaling at the MHD scales and its restoration at the flanks and in the inner parts of the MSH (far enough from the BS). At the kinetic scales, spectra tend to steepen in the dayside MSH and recover the shape observed in the SW at the flanks.

For the compressed SW streams like CIR and Sheath, substantial deviation from the Kolmogorov scaling occurs at the MHD scales throughout the dayside MSH, with the spectra being generally shallower than $f^{-1}$. This may be a result of wave processes dominating the cascade in the dayside MSH for specific SW features. At the kinetic scales, spectra usually exhibit substantial steepening in the dayside MSH. Moreover, there is a linear dependency between the slope value in the dayside MSH and the SW; unusually steep spectra in the SW that are typical for compressed SW streams are then followed by untypically steep spectra in the dayside MSH. A similar effect was found in the downstream regions of the interplanetary shocks [65]. A case study [85] also demonstrated that spectra downstream of the BS are steeper than upstream for different SW types.

At the flank MSH, the slopes of the spectra at the kinetic scales are generally close to the values $-7/3$ or $-8/3$ predicted in theoretical frameworks [16,17]. Note that typically in the SW [28–31] and throughout the MSH [24–26], spectra follow the $f^{-2.8}$ law at the kinetic scales. The present study demonstrates that there is no relation between the slopes at the flank and in the upstream dayside MSH. However, there is a clear tendency to observe spectra with a slope close to $-7/3$ during the compressed SW streams. This implies a specific form of cascade development for the compressive component of the fluctuations when the plasma moves away from the BS.

For the disturbed SW, streams referred to as ICMEs (MC and Ejecta), the modification of the spectra at the MHD scales seems not to be such dramatical and results in spectra with $-1$ slope in the dayside MSH, which is typical for this region [36]. The main difference between the compressed SW flows and ICMEs lies in the kinetic scales. For compressed SW, the spectra steepen in the dayside MSH and then untypically flatten at the flanks assuming the pump of the additional energy to the kinetic-scale fluctuations when the plasma moves

away from the BS. For the ICMEs, the substantial steepening of the spectra occurs in the dayside MSH with the further restoration of the spectra toward the flanks, with the spectral slopes having values close to those observed in the SW.

Thus, the study demonstrates different preliminary scenarios of the turbulence evolution inside the MSH for specific SW conditions. The question is if the differences in local conditions result in specific spectral shapes or if this is a complex effect of the specific evolution of plasma turbulence for the SW streams of different Sun sources.

Though the twelve cases considered here cannot be treated to obtain a reliable statistical result to answer this question, some preliminary dependencies can be obtained. Values of the spectral slopes at the MHD scales and the kinetic scales have been considered versus the local plasma and magnetic field parameters such as density, velocity, magnetic field magnitude, $B_z$ component, ion temperature, the angle between velocity and magnetic field vectors, Taylor ratio $V/V_A$ and ion characteristic frequencies as well as initial power of the fluctuations (power spectral density) at the MHD scales. Figure 7 demonstrates those dependencies for the dayside MSH where the absolute value of Pearson's correlation coefficient is 0.5 and higher. The type of SW for each point is indicated in the legend by different colors. Note that for one of the considered cases (9 July 2014), Spektr-R measurements were used in the dayside MSH, so the magnetic field measurements and some of the calculated parameters were not available for this case. For this reason, only 11 points are present in Figure 7c. In addition, the slope $P_1$ cannot be calculated reliably for two spectra in the dayside MSH as mentioned in the previous section. This is the reason for the reduced number of points in panels (d,e).

Panel (a–c) presents the tendency to observe steeper spectra at the kinetic scales in the dayside MSH with increasing density and characteristic frequencies, which include density. On the other hand, the correlation coefficients of the slope $P_2$ versus magnetic field magnitude and ion temperature (not shown) were estimated as $R = -0.37$ and $R = -0.1$, i.e., the correlation was low. Thus, features of the turbulence in the dayside MSH seem to be dependent on the level of the dayside MSH compression, which results in the dependence of the $P_2$ on the ion density and subsequent dependence on the gyrostructure frequency $F_G$ and inertial length frequency $F_L$. The tendency of steepening the density and velocity fluctuation spectra at ion kinetic scales with the increasing compression level has been demonstrated recently with the help of MMS data [52]. Note that according to Figure 7a, the slope $P_2$ decreases with ion density regardless of the SW type. Thus, substantial steepening in the dayside is likely to be due to the increased compression during SW of types Sheath and CIR rather than due to any other specific features of these streams.

Panels (d,e) show dependencies of the slope $P_1$ at the MHD scales vs. the angle between velocity and magnetic field vectors $\alpha(V,B)$ and versus the angle $\theta_{BN}$. The spectra with untypical slopes at the MHD scales ($|P_1| < 1$) are likely to be observed when $\alpha(V,B)$ is around 90°. If the Taylor hypothesis is assumed, then cases of $\alpha(V,B) = 90°$ consider the fluctuations with k normal to B, and cases with V parallel to B refer to the fluctuations with k parallel to B. Two types of instabilities that dominate the MSH behind the quasi-perpendicular BS differ by the wave vector component: for mirror modes, k is mostly perpendicular to the B, while for AIC waves, k is mostly parallel to B [9]. Thus, the tendency shown in Figure 7d may imply that fluctuations that present substantial deviation from the Kolmogorov scaling are of the mirror mode nature. However, this point needs more accurate analysis. There is also a tendency to observe spectra with Kolmogorov scaling when the $\theta_{BN}$ angle decreases, i.e., when the transition to quasi-parallel BS occurs. This result corresponds well with previously reported [19,22]. The study [52] demonstrated the shallower ion density fluctuation spectra at the MHD scales with increasing levels of compression. The present study has not found any particular dependence of the MHD-scale slope on the level of compression. However, for the highly compressed SW streams, the spectra in the dayside MSH always are shallower than the Kolmogorov spectrum. They are often characterized by the slope $|P_1| < 1$, which corresponds well with previous results [38].

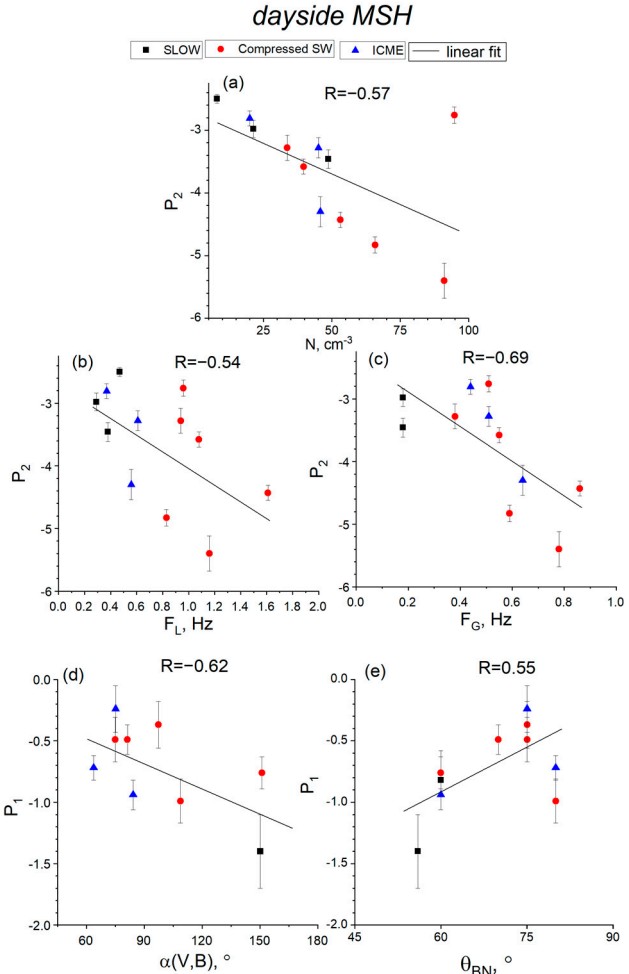

**Figure 7.** Slope $P_2$ at the kinetic scales in the dayside MSH vs. ion density (**a**), inertial length frequency $F_L$ (**b**), gyrostructure frequency $F_G$ (**c**); slope $P_1$ at the MHD scales in the dayside MSH vs. angle between V and B vectors (**d**), $\theta_{BN}$ angle (**e**).

Interestingly, none of the considered dependencies at the flank gave a high enough correlation coefficient. However, according to the results described in Section 4, there are clear differences in spectral features for compressed SW streams and other considered types. This suggests specific evolution of the turbulent cascade involved in the compressed streams while they interact with the BS and propagate through the MSH.

Generally, the results of the analysis of the compressive component of the turbulent cascade can be summarized as follows:

1. During the slow undisturbed SW streams, the fluctuation spectrum may be modified in different ways at the MHD scales in the dayside MSH and tends to restore Kolmogorov scaling at the flanks; at the kinetic scales, slight steepening of the spectrum occurs at the dayside MSH with the restoration of the initial SW slope at the flanks;
2. During disturbed SW streams, substantial deviation from the Kolmogorov scaling occurs at the MHD scales in the dayside MSH; at the flanks, the Kolmogorov scaling is typically restored except for the cases in particular vicinity of the BS;
3. Steepening of the spectra behind the BS occurs for all types of the SW; the compressed SW streams are characterized by slightly steeper spectra than typically observed in the SW, which become untypically steep in the dayside MSH;
4. Steepening of the spectra in the dayside MSH at the kinetic scales seems to be controlled by the level of plasma compression in the dayside MSH, with steeper spectra presenting during the more compressed plasma;

5. Turbulent cascade embedded to the compressed SW streams evolves in the MSH in a specific way that results in untypically flat spectra at the flanks; specific redistribution of the energy through the cascade when plasma propagates away from the BS is likely to be a feature of these SW streams.

Presented several examples of the three-point measurements of turbulent compressive fluctuations reveal the different scenarios of turbulence modification at the quasi-perpendicular BS and throughout the MSH behind it. Combining present results with the achievements of previous studies, the turbulence evolution at the BS and inside the MSH can be pictured as preliminary. Upstream of the BS, the turbulent spectrum follows Kolmogorov scaling at the MHD scales and the power law with the exponent depending on the background conditions (including the type of the SW flow) at frequencies higher than the ion spectral break. At the BS, the Kolmogorov scaling locally ruins, and the inertial range in the cascade is absent. Simultaneously, the fluctuations around ion scales are dominated by instabilities, resulting from high-temperature anisotropy behind the BS. At the considered scales, enhanced dissipation due to the crossing of the shock operates. When plasma moves away from the BS, these effects decrease. The distance at which the cascade restores its initial SW shape depends on the SW type: for undisturbed slow SW, the BS effect occurs in the vicinity of the shock front, while for ICMEs or compressed SW regions, the BS effect covers whole dayside MSH. When plasma propagates to the MSH flanks, the BS influence on the turbulent cascade can be observed only in the vicinity of the BS front except for the cases of compressed SW streams. Significant compression in the SW may result in additional energy of compressive fluctuations, which redistributes over the cascade toward the smaller scales and results in flatter spectra of the compressive fluctuations at the MSH flanks. Further comprehensive analysis with the help of a larger dataset of different spacecraft measurements is required to obtain a full picture of turbulence evolution in the MSH.

**Supplementary Materials:** The following supporting information can be downloaded at: https://www.mdpi.com/article/10.3390/universe8120611/s1: Table S1, Considered cases, and characteristic background parameters.

**Author Contributions:** Conceptualization, G.N.Z.; Data curation, M.O.R.; Investigation, L.S.R.; Methodology, M.O.R.; Project administration, Y.I.Y.; Software, L.S.R. and M.O.R.; Supervision, M.O.R. and Y.I.Y.; Validation, G.N.Z. and Y.I.Y.; Visualization, L.S.R.; Writing—original draft, L.S.R. All authors have read and agreed to the published version of the manuscript.

**Funding:** The work was supported by the Russian Science Foundation, grant 22-12-00227.

**Data Availability Statement:** Wind and Themis data can be found at https://cdaweb.gsfc.nasa.gov/ (accessed on 1 August 2022); the Spektr-R data (BMSW instrument) can be found at http://catalog-sw-msh.plasma-f.cosmos.ru/ (accessed on 1 August 2022), data with the highest time resolution are available on request.

**Conflicts of Interest:** The authors declare no conflict of interest.

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
