# Peer review of "Large-Scale Solar Wind Phenomena Affecting the Turbulent Cascade Evolution behind the Quasi-Perpendicular Bow Shock"

_universe, doi:10.3390/universe8120611_

Round 1

Reviewer 1 Report

The turbulent cascade behind the bow shock is an interesting topic, and this manuscript investigated the influences of the solar wind parameters on the dynamics of turbulence behind the bow shock. Some new findings have been obtained, and the manuscript is well-written. I think that the manuscript has the potential to be published, however, I still have several comments.

1.       Different from the turbulence in the solar wind, the turbulence in the magnetosheath may be influenced from the compression of the bow shock, the boundary from the magnetopause , the solar wind parameters etc. Although it is difficult to analyze what is the exact role of the influences, it still need more discussion on these influences.

2.       On line 143-151, the authors used the shape of spectra of ion flux fluctuations represent that of normalized magnetic field amplitude, because the previous observations found that they have the similar spectra. Some physical discussion is necessary to demonstrate that they have the similar shape of spectra.

3.       On line 37, more references are necessary to demonstrate that plasma waves etc. in the magnetosheath can be incorporated into turbulence, like Lucek et al., Space Sci. Rev., 118, 95-152, 2005.

4.       On line 39-41, the following paper [Lu et al., Geophys. Res. Lett., 47, e2019GL085661, 2020] also described a global hybrid simulation model of the near-earth plasma, and should be cited.

Reviewer 2 Report

The aim of the paper is simultaneous plasma observation in three points of the near-Earth’s space: in the upstream solar wind, in the dayside magnetosheath behind quasi-perpendicular bow-shock and at the magnetosheath flank. The study focuses on the range of scales around the ion characteristic scales. Measurements of WIND, Themis and ion flux from Spektr-R spacecraft have been used.

The approach is interesting and not easy. I encourage the authors to answer the below comments.

General comments

In the introduction of the paper, the reader understand that any magnetosheath configuration will be considered, but on page 6, we read that the authors will focus on MSH behind Q_perp BS. This should be said from very beginning,  i.e. in the title of the paper, in the abstract and in the introduction.

It will be helpful for the reader if the introduction gives more deep overview of the fluctuations in the Earth's magnetosheath research. Indeed, the present discussion is superficial. For example, term 'variability' is not clear for the reader. Please specify the exact results of the papers discussed.
As far as the paper is focused on Q_perp MSH, the introduction must include a comparison between Q_perp and Q_|| magnetosheaths, or focus mostly on the results obtained for the Q_perp one. What are the open questions of this research ?

The last section lacks interpretation or an attempt for it.
To show time series of the data used to calculate spectra will be very useful and may help with the interpretation of the results.

Please add in the introduction some old studied on magnetosheath fluctuations and turbulence, which can be important for the present study:

Anderson et al. (1994), Magnetic spectral signatures in the Earth’s magnetosheath and plasma depletion layer, J. Geophys. Res., 99, 5877–5891

Lacombe, C., “Density and magnetic field fluctuations observed by ISEE 1-2 in the quiet magnetosheath”, Annales Geophysicae, vol. 13, no. 4, pp. 343–357, 1995. doi:10.1007/s00585-995-0343-1.

Czaykowska, A., Bauer, T. M., Treumann, R. A., and Baumjohann, W., “Magnetic field fluctuations across the Earth's bow shock”, Annales Geophysicae, vol. 19, no. 3, pp. 275–287, 2001. doi:10.5194/angeo-19-275-2001.

Czaykowska, A., Bauer, T. M., Treumann, R. A., and Baumjohann, W., “Mirror waves downstream of the quasi-perpendicular bow shock”, Journal of Geophysical Research, vol. 103, no. A3, pp. 4747–4752, 1998. doi:10.1029/97JA03245.

Hellinger, P., Trávníček, P., Mangeney, A., and Grappin, R., “Hybrid simulations of the magnetosheath compression: Marginal stability path”, Geophysical Research Letters, vol. 30, no. 18, 2003. doi:10.1029/2003GL017855.
https://ui.adsabs.harvard.edu/abs/2003GeoRL..30.1959H/abstract

Alexandrova, O., Mangeney, A., Maksimovic, M., Cornilleau-Wehrlin, N., Bosqued, J.-M., and André, M., “Alfvén vortex filaments observed in magnetosheath downstream of a quasi-perpendicular bow shock”, Journal of Geophysical Research (Space Physics), vol. 111, no. A12, 2006. doi:10.1029/2006JA011934.

Alexandrova, O., “Solar wind vs magnetosheath turbulence and Alfvén vortices”, Nonlinear Processes in Geophysics, vol. 15, no. 1, pp. 95–108, 2008. doi:10.5194/npg-15-95-2008.

Matteini, L., et al., “Electric and magnetic spectra from MHD to electron scales in the magnetosheath”, Monthly Notices of the Royal Astronomical Society, vol. 466, no. 1, pp. 945–951, 2017. doi:10.1093/mnras/stw3163.

and the 1st paper on density spectrum with flattening at ion scales:
    Celnikier, L. M., Harvey, C. C., Jegou, R., Moricet, P., and Kemp, M., “A determination of the electron density fluctuation spectrum in the solar wind, using the ISEE propagation experiment”, <i>Astronomy and Astrophysics</i>, vol. 126, no. 2, pp. 293–298, 1983.

Also, different solar wind types considered in the study:
1) undisturbed slow solar wind  
2) disturbed compressed SW streams - Sheath, CIR;
3) disturbed SW associated with ICMEs – MC, Ejecta
must be mentioned in the abstract and introduction.

The terms 'above' and 'below' are not clear. Will be easier for the reader if you use 'at frequencies higher (lower) then ion scales'.

Specific comments:

Page 2 : please give the values of spectral indices all over the discussion, to be more specific.
It is clear that the BS geometry, Q|| or Qperp, will define the magnetosheath fluctuations. Please comparer these two configurations in the introduction.

page 2: 'Development of the turbulent cascade after energy injection at the BS...' -- This statement seems for me under debate...  is it well established that the energy injection is at the bow-shock ? why not just the SW cascade more or less modified by the shock ?

Page 3
Reference [32] is not avalible on-line, can you resume in more details the results ?

line 114-115 : 'specific scenarios of turbulence'
strange... turbulence is known to have a universal spectrum far from characteristic scales. Please, specify what do you mean here.

line 120 : 'Fast measurements..' -- it is relative.
line 132-133: please give here the exact time resolution
lines 139-140: break => ion scales; fast => give time resolution for different instruments.
lines 145-146: "Fluctuations of ion flux value are believed to represent fluctuations of density [41, 42]."-- why 'believed' ?
Are there theoretical work done to show the relation between ion flux and density ? Please give more details and references.

Page 4
lines 174-183: very interesting. Plasma propagation in the magnetosheath is not radial and seems difficult to do this identification. Please, give more details on the path estimates between 3 points. A simple sketch may help to explain to the reader the method and the path.

lines 195-196: 'results from its significantly higher propagation speed compared to the ambient plasma'
-- not clear, give V measurements

lines 197-200: '... there are a lot of similar small-scale features which do not match in time for the selected shift between the spacecraft data rows. This results from different speeds of propagation of small-scale structures'
-- or a fast time evolution of small scale structures... please discuss this option.
Interesting that for shift the data you use the time shift which do not put together the sharp discontinuity around 23:00. Please comment.

page 5
Discussion of Fig.1(e): please add here
- the ion scales frequencies
- frequency ranges in Hz where the fitting is done
- a discussion on the fact that a clear break is not observed, instead we observe a smooth transition from one power-law to another one at ion scales.

page 6
lines 234-235: 'For the presented case the spacecraft were separated by more than 250 RE and still the spectra demonstrated similar shape and features'
-- The MHD range shows different spectral indices. Taking in to account the errors, still the indices are not overlapped... Is it possible to include density spectra from Wind to confirme the MHD range similarities ?

line 239: are the scales really sub-ion ? the reader do not see here ion scales indicated ...
Please add to the Fig.1e f_ci and doppler shifted ion larger radius and ion inertial length.

lines 246-248: 'Thus, we suggest that spectra of magnetic field magnitude fluctuations on board the WIND can be successfully compared to the ion flux fluctuation spectra on board the Spektr-R spacecraft'
-- will be more convincing for the reader if PSD(density_on_wind) was shown as well.

line 266: 'database of the BMSW' -- what is this ?
line 272: 'high value of the correlation coefficient' -- what does it mean ? please give the threshold of the correlation coefficient you use for this analysis.

line 276: there is a strange symbole/letter after 'val'
line 280: MC -- do you mean 'magnetic cloud' ? please specify; the same for CIR, ICME and Ejecta -- not clear for the reader.

page 7

lines 286-287: this information is important and must be mentioned in the abstract together with the fact that only Q_perp BS and MSH behind are considered.

lines 301-302: 'The values of the V/VA are listed in Table 1 of the 301 Supplementary materials.'  -- why in Supp. Mat.? it seems important to give this informations in the main body of the paper.

line 307: add here a reference on whistlers in the solar wind by Lacombe et al. 2014, APJ.

line 311: Panel (f) instead of (e), right ?
line 323: 'while at the flank the parameters are closer to the SW observations' -- not clear from Fig.2, please give more indications. which parameter on SpR is close to SW ?

line 325: 'The shaded area denotes'
-- please give here the time on one of the s/c, and then indicated the time delays for the others used. In order that anybody can re-do the analysis and verify the results.

page 8
line 339: please include Table 1 in the paper.
line 346: 'close to -8/3 power exponent'  
-2.5 is not -2.66, please say just what do you observe. Mention as well, that the range of frequencies for the fitting is very small, from 2 to 5 Hz, so there is no so much meaning in a deep interpretation of -2.5 obtained here.

lines 346-348: 'Behind the BS in the dayside MSH (blue line) the bump in the spectrum is formed at frequency ~0.04 Hz, which is close to the ion cyclotron frequency.'
-- what is the origine of this bump? Probably Mirror instability is at work that is usually the case in the Q_perp MSH for plasma beta>2. For beta<1, AIC is expected. For the intermediate beta, both instabilities can take place.

lines 349-350: 'At the kinetic scales the spectrum is substantially steeper than in the SW and than observed usually in the SW and in the MSH'.
-- This spectral index may be due to the bump at lower frequencies. as just a continuation, that we see from Fig2. Such steep spectra can be also a result of the presence of coherent structures of the vortex type, see Alexandrova et al. 2006 JGR, where f^-3 is observed at kinetic cales and 2008 NPG, where f^-4 is observed.

lines 350-353: 'At the MHS flank the spectrum is shallower at the MHD scales than observed typically for the developed turbulence in the undisturbed SW characterized by the Kolmogorov scaling. ...'
-- the range of frequencies lower then the bump and higher are very short to give any conclusion on the shape. Please, do not over interpret the observations.
Discuss as well the presence of the bump in ion flux spectrum. Does it mean that Mirror instability continues to be active in the flanks ?

line 354: 'Also, bump in the spectrum occurs at frequencies ~0.3 Hz'  
--will be more appropriate to say "around 0.3 Hz, and it covers ..."-- give here the range of the covered frequencies.

lines 354-358: 'Thus, for the presented case of undisturbed slow SW flow the turbulent spectrum has typical features in the SW, steepens at the kinetic scales when plasma crosses the BS and recover its shape at the kinetic scales when plasma moves toward the flanks; at the MHD scales spectrum at the flank MSH in the vicinity of the BS deviates from Kolmogorov scaling.'

-- I do not agree with these conclusions. From Fig.2 one concludes that for the typical slow SW, the fluctuations in the MSH are defined by local T_perp/T_|| anisotropy instabilities, as is usually observed in the Q_perp MSH, see Schwartz, Burgess, Moses, 1996.
The spectral indices fitted on frequency ranges much smaller than a decade are not very informative.
Spectral indices close to -1 can be interpreted as a superposition of incoherent waves [Horbury 2005], that is usually the case at low frequencies in the Q_perp MSH, see the introduction of [Alexandrova 2008 NPG] paper.

page 9, line 362: 'However, properties of this wave process are out of the scope of present study.'
-- it is unfortunate, because without checking the wave mode nature, it is difficult to give any conclusion on the spectral analysis. I encourage the authors to check time series of |B| at least.

page 10, lines 439-453: 'Note that the slope -5.4 is highly untypical for the ion kinetic scales in the MSH though it 439 is often observed at smaller electron scales at which further steepening occurs (statistics 440 can be found in [21]).'
As one can see from Fig.4, the -5.4 spectrum is around ion scales (where any turbulent model can be applied because we are not far away from the characteristic scales), and it seems just to be the continuation of a spectral bump at 0.1-0.5 Hz. The same can be said about the statistics of ref. [21], where the spectral bumps due to whistler wves have been fitted with a power-law, and there is no meaning in it.
Here, I encourage the authors, to show to the reader time series of data on which the FFT have been applied.
It will be great to have these plots for other time intervals studied in the paper.

 line 482-583:  "The spectrum in the SW is somewhat steeper than Kolmogorov’s..."
 -- please give the value of the spectral index in the text so that the reader can follow. Do you mean f^-1.68 or f^-1.89 ?
Is there enough frequency range (one decade and more) before arriving at ion scales to determine the MHD range scaling ?

line 485-486: spectrum in the dayside MSH with flattening at large scales and -4 at small scales can be a spectrum of a network of Alfven vortices, see Alexandrova 2008 NPG.

lines 572-573: 'Note, that typically in the SW and throughout the MSH spectra follow f -2.8 law at the kinetic scales [63, 64, 21, 573 22].'
-- add here references on Alexandrova et al. 2012 APJ & 2021 PRE for the SW and Matteini et al. 2017 for the MSH.

page 15
caption of Figure 7: i'm not sure that the caption describes exactly what is shown. please, re-write the caption.

lines 627-628: "The spectra with untypical slopes at the MHD scales (|P1|<1) are likely to be observed when the angle between V and B tends to zero."
-- this is a very important result, which indicates that the untypical slopes are for k_|| spectra.  
it is consistent with observations of a mixture of quasi-parallel AIC waves for example.
Please discuss these ideas (the link betweeb Theta_BV and k-anisotropy and possible AIC behind the spectra) in the text.

lines 629-630: 'There is also a tendency to observe spectra with Kolmogorov scaling when θBN angle decreases i.e. when transition to quasi-parallel BS occurs.'
 -- it is a well known result that behind a Quasi-|| shock, the sw turbulence is just amplified.

lines 633-636:
'Present study has not found any particular dependence of the MHD-scale slope on the level of compression. However, for the highly compressed SW streams the spectra in the dayside MSH always are shallower than Kolmogorov spectrum and are often characterized with the slope |P1|<1.'
-- Please mention that it is in agreement with previous observations by Czaykowska et al. 2001.

Page 16
The interpretation or a tentative of it is needed.

Reviewer 3 Report

This paper presented several examples of multipoint observations of turbulent fluctuations in the solar wind, in the dayside magnetosheath and at the flanks, trying to reveal the differences in the development of the turbulent cascade in the MSH during SW streams with specific properties. This paper is interesting and well-written. It should be considered for publication after some modifications. Please see my comments below.

1.     Figure 2(f): It should be Themis-D instead of Themis-A.

2.     Are the 12 cases examined in this paper all of the events that meet the data selection criteria outlined in section 2? It seems strange to me that the number of cases in the slow solar wind is lower than the number of cases in CIRs.

3.     The ICME intervals are divided into MCs and ejecta in section 2.6, but they are not treated separately in the following. I would advise against distinguishing between them.

4.     It seems that compression is very important for the spectra shape evolution. So, what are the speeds of these ICMEs? Are they fast ICMEs or slow ICMEs?

5.     In lines 472-476, you mentioned that significant steepening of the spectra at the kinetic scales is observed in the dayside MSH, and that steeper spectra in the SW are followed by steeper spectra behind the BS. That is not appropriate to say here, in my opinion. Because the previous section only provided one example. Furthermore, as shown in Figure 6, there are no obvious signs of steeper spectra in the SW followed by more significant steepening behind the BS.

6.     Line 606, it is insufficient to assert that there are clear dependencies of the P2 slope in the dayside MSH on ion density and characteristic frequencies, given that the coefficients are only 0.57,0.54, and 0.69 and the sample size is only 12.

7.     In figure 6 and figure 7, not all cases are shown in the figure, please check carefully.

Round 2

Reviewer 1 Report

I think that the authors have revised the manuscript properly, and it is now ready to be published.

Author Response

Thank you for careful revision and interest to our manuscript.

Reviewer 2 Report

Report-2 on Article
Title: Large-scale solar wind phenomena affecting the turbulent cascade evolution behind the bow shock
Journal: Universe

The authors addressed most or the comments. Last improvements are needed before the publications:

- in the abstarct there is no conclusion what happens for the solar wind with CME, please add a sentences on it. 

- lines 56-59 : 'Comprehensive experimental studies of the turbulence features in the MSH at the scales around and below the proton gyroradius have been performed since the launch of the Cluster mission (see reviews [17, 18]). Recent measurements by MMS helped to move on and consider even smaller scales - up to electron scales (e.g. [19-21]).'

-- It is not exact, not with MMS that we start to look at electron scaled. 
It is with Cluster, that the first studies have been done at electron scales, see Mangeney et al. 2006, Lacombe et al. 2006, Matteini et al. 2017. 

lines 62-64: 'These properties are close to what is observed typically for the SW in the pristine solar wind without large-scale disturbances [26, 27].'
-- please cite here as well first case study of the observation of -2.8 spectrum in the pristine solar wind at sub-ion scales [Denskat et al. 1983 Journal of Geophysics], and first statistical studies indicating generality of this power-law at sub-ion scales [Alexandrova et al. 2009 PRL, 2012 APJ]. 

lines 65-66: 'However, values of the power exponents have wide distributions and are supposed to depend on a number of factors described below.'
-- it is not the case for sub-ion scales, see statistical study Alexandrova et al. 2012 and 2021,
and in particular, histogrammes of spectrum indices between ion and electron scales, Alexandrova et al. 2012, Figure 5(a), -- distribution of \alpha_1 is very narrow. 
The variability is present at ion scales, i.e, at [0.1-3] Hz, where spectral index may vary between -4 and -2, see Smith et al. 2006. The paper of Lion et al. 2016 is about MHD-ion scales and not on sub-ion scales. 

lines 67-68: 'Moreover, usually the pure turbulent spectrum is superimposed by spectra of other processes like coherent structures or waves that results in various spectral features (e.g. [29]).'
-- coherent structures (or intermittency) are important part of the cascade, they are not superimposed on turbulence, but a part of turbulence. However, quasi-linear instabilities are indeed superimposed on background turbulence and thus may create breaks (or bumps) in the spectra. 

line 98: 'Alfven vortices [43, 44] at ion scales which are not typical to the SW plasma.' 
-- More recently, it was shown that Alfven vortices are also present in the solar wind at the end of the inertial range-at ion scales, see [Lion et al., 2016, Roberts et al. 2016, Perrone et al. 2016, 2017]. 

lines 137-139:
'At the kinetic scales [50] the -2.8 index is typical for slow undisturbed SW while for the compressed plasma flows this index usually has higher absolute values -3.2.'
-- please specify the range of frequencies with respect the smallest ion scales. Normally, independently on the solar wind type and plasma parameters, at scales smaller then all ion scales (f_ci, f_rho_i and f_lambda_i), the spectral index is -2.8.
It is possible that -3.2 is observed between f_ci and f_\rho_i or f_\lambda_i, where f_rho_i is Doppller shifted ion Larmor radius and f_lambda_i is Doppler shifted ion inertial scale. 

lines 146-148: "When compressed Sheath stream interacts with the MSH some of the MSH small-scale fluctuations can preserve from the SW while some of them originate inside the MSH [52]."
-- It is not evident at all. It is possible that at shock all scales of the cascade are mixed-up and then downstream the cascade forms again. As far as kinetic scales characteristic times are very short and plasma turbulence is universal, we observe the same spectral index very close to the shock. But it is not the proof that all these fluctuations preserve from the SW, spectral index is not a proof of causality. 

line 180: 'The scales of interest 180 are around ion scales,' - very good, now it is clear !

page 6, Fig.1, caption '(e) comparison of the magnetic field magnitude (red line) and ion flux (blue line) spectra at WIND (red line) and ion flux spectrum at Spektr-R (black line) for the analyzed case.' 
-- y-axis units are not clear. Probably the spectra are normalised... please explain the normalization  in the text and give it in the figure as y-axis title and or in the caption 

Figure 1(e): the PSD need to be more thick in order to see them in the printed version. 

line 415 : 'of coherent structures like Alfven vortices [43, 44].' please add here that it can be also the continuation of the instability bumps visible at lower frequencies. 

line 549:'The spectrum in the SW is somewhat steeper than Kolmogorov’s at the MHD scales with the slope -1.89±0.13' 
-- this is close to -2, that is a spectrum of a discontinuity. Please comment. May be you already include shock in calculation of the PSD ? please check. 

line 746: 'at the subion scales' 
- it seems that the analysis presented in the paper is on ion scales (as was said in line 180) and not so much on sub-ion scales. So, please, correct your discussion in the conclusions. 

line 748: 'the cascade' please change it to 'the fluctuations'
because it is not sure that ion instabilities are the part of the cascade.
